# An All-Solid-State Flexible Supercapacitor Based on MXene/MSA Ionogel and Polyaniline Electrode with Wide Temperature Range, High Stability, and High Energy Density

**DOI:** 10.3390/molecules28041554

**Published:** 2023-02-06

**Authors:** Feng Pan, Shuang Wang, Zhipeng Yong, Xiaodong Wang, Chenglong Li, Dan Liang, Xiaorui Wang, Han Sun, Yinghe Cui, Zhe Wang

**Affiliations:** 1College of Chemical Engineering, Changchun University of Technology, Changchun 130012, China; 2Advanced Institute of Materials Science, Changchun University of Technology, Changchun 130012, China

**Keywords:** ionogel, Mxene, high temperature, flexible supercapacitors

## Abstract

In this study, an ionogel electrolyte (PAIM-X) consisting of 1-vinyl-3-methylimidazole bis (trifluoromethyl sulfonyl) imide ([VMIM][TFSI]), Polyacrylamide (PAAm), and MXene were prepared. The conductivity of PAIM-X and integral area of the voltammetric curve of the supercapacitor (PAIMSC) were improved by adding MXene. The addition of [VMIM][TFSI] enhanced the conductivity and applicable temperature of the ionogel electrolyte. At 90 °C, the conductivity of PAIM-4 can reach 36.4 mS/cm. In addition, spherical polyaniline with good electrochemical properties was synthesized and coated on graphite paper as an active substance. An all-solid-state supercapacitor was composed of PAIM-4, polyaniline electrode with 1.2 V potential window, pseudo-capacitors and high quality capacitors. The solvent 1-ethyl-3-methylimidazolium bis (trifluoromethyl sulfonyl imide) ([EMIM][TFSI]) and methanesulfonic acid (MSA) were introduced into the ionogel to promote the redox reaction of polyaniline (PANI). The mass specific capacitance of PAIMSC was 204.6 F/g and its energy density could reach 40.92 Wh/kg, which shows great potential for practical application at high temperature. The device had good rate performance and cycle performance, and its capacitance retention rate was still 91.56% after 10,000 cycles. In addition, the supercapacitor can work within the temperature range of −20 °C to 90 °C. These excellent electrochemical properties indicate that PAAm/IL/Mxene-X has broad application space and prospect.

## 1. Introduction

With the continuous progress of science and technology, new energy equipment has come into being. SCs have been widely used in various fields due to their excellent high energy density, good long-term stability, and environmental friendliness [1,2]. However, the low operating voltage and capacitance of SCs limit their applications. Liquid electrolytes with wide potential windows are often used [3]. However, they have risks of explosion and fire due to leakage. In order to avoid these situations, solid electrolytes have been developed to operate over a wide temperature range and high potential windows [4]. Solid-state supercapacitors consist of electrodes and electrolytes. The electrode material is usually made up of one or more of metal oxides, conductive polymers, or carbon-based materials. Electrolytes are usually composed of gel electrolytes and solid polymer electrolytes. Pure solid polymer electrolytes have poor interface contact with electrodes and low ionic conductivity. This disadvantage limits the energy density of supercapacitors and shortens their application range. Gel electrolytes are widely used because of their good flexibility and good contact with electrodes. By enlarging the electrochemical window of the supercapacitor or reducing the resistance of the gel electrolyte, the energy density of the electrolyte can be increased while ensuring the power density. Compared with hydrogel electrolytes, ionogel electrolytes (IGEs) using ionic liquids as solvents and monomers just meet these two conditions [5,6]. MXene is obtained by selective etching of the “A” metal element in the MAX phase. The MAX phase is a group of ternary carbides and nitrides, where “M” is the transition metal and “X” is generally C. The general formula of MXene is M_n+1_X_n_T_x_, where T_x_ represents the surface -O, -OH, -F, -Cl, and other functional groups of MXene. These abundant surface groups make it possible to combine with graphene, polymers, metal ions, etc., to form gel electrolytes. Two-dimensional or three-dimensional network structures were formed by physical or chemical cross-linking of the cross-linking sites of Mxene nanoplates and gel systems. This not only avoids MXene’s self-accumulation, but also provides a channel for gel ion transport, which is widely used in batteries and supercapacitors [7,8].

Patrice Simon et al. further immersed the hydrogel film into the electrolyte of [VMIM][TFSI]. The ionogel films exhibited a 70 F/g capacitance at a rate of 20 mV/s [9]. Udo Kragl et al. found that *N*,*N*′-methylene bisacrylamide and vinyl can form polyionic liquid hydrogels in water. The prepared hydrogel has good thermal stability and mechanical properties. Therefore, the gel polymer electrolyte was prepared with [VMIM][TFSI] [10]. Yohan Dall’Agnese et al. found that with MSA/PVA polyvinyl acetate as the electrolyte and MXene as the electrode of the supercapacitor has the advantages of high cycle life and low temperature resistance [11]. Tifeng Jiao et al. found that by adding gelatin-modified MXene to the Polyacrylamide (PAAm) hydrogel, the tensile strength increased significantly (1100%) and the tensile strength increased significantly (430 kPa). [12]. Up to now, there have been few studies of all-solid supercapacitors based on ionic liquid gel electrolytes. Poly(3,4-ethylenedioxythiophene) (PEDOT) [13], Polypyrrole (PPy) [14], and Polyaniline (PANI) [15] are widely used as conductive polymer materials for supercapacitor electrodes because they produce the Faraday oxidation–reduction process of false capacitance and have high specific capacitance. The conductive polymer PANI undergoes the process of doping and redox of undoping electrolyte ions to store electric energy. Because of its high storage capacity, high doping/undoping rate, and low cost, it is considered one of the most studied conductive polymers [16,17].

In this paper, [VMIM][TFSI] and PAAm were used as monomers, and MXene was used as an inorganic filler to provide an ion transport channel for ionogel. In addition, methanesulfonic acid and [EMIM][TFSI] have also been used innovatively as solvents for ionogel electrolytes. As [VMIM][TFSI] and [EMIM][TFSI] have similar structures, excellent electrochemical performance, and good compatibility with methanesulfonic acid, the prepared ionogel has higher conductivity and wider application temperature. Under ice bath conditions, spherical polyaniline was prepared and coated on graphite paper as an active substance. The electrochemical properties of PAIMSC were measured at different temperatures by using ionogel and polyaniline electrodes to assemble a supercapacitor.

## 2. Results and Discussion

### 2.1. Synthesis and Mechanism of PAIM-X Ionogel Electrolyte

Figure 1a,b are two digital photos of ionogel. The gel without MXene in Figure 1a is colorless and transparent, but it becomes a uniform black ionogel after adding MXene. In Figure 1b, the prepared PAIM ionogel has the ability to resist twisting and stretching, which indicates that it has excellent flexibility [18]. The elastic modulus of the MXene proportional gel with different compositions was determined by tensile test. As shown in the Figure 1c, the conductivity of PAIM-X ionogel electrolyte is positively correlated with temperature. With the increase in temperature, the ion migration is accelerated along with the thermal motion of polymer molecular chain [19,20]. At the same temperature, the conductivity of PAIM-4 ionogel electrolyte is the highest, and the conductivity of PAIM-4 is 36.4 mS/cm at 90 °C. It was attributed to the -CONH_2_ group of PAAm forming hydrogen bonds with the -OH and -F groups on the surface of MXene, forming an interconnected network between the monomers. It provided a shorter path for protons and improved the conductivity of the gel [21]. However, when the content of MXene reached 5 mg/mL, the conductivity of the ionogel decreased. This is due to the low conductivity of MXene itself and the low charge transfer rate caused by excessive crosslinking. By analyzing the proton conductivity and mechanical properties of the gel, the optimum addition of MXene was 4 mg/mL. In order to verify the good conductivity of PAIM-X at high temperature, we measured the proton conductivity of PAIM-X from 10 °C to 90 °C. As shown in Figure 1d, at the same tensile rate, with the increase in MXene content in gel, the elastic modulus increases and the tensile strain decreases. When the content of MXene is 4 mg/mL, the elastic modulus of MXene gel is 33.4 kPa, which is about twice that of the gel without MXene. This is because the cross-linking effect of MXene increases the tensile strength and decreases the elongation at break. When the content of MXene is above 5 mg/mL, the mechanical properties of MXene decrease due to the aggregation of MXene. This is consistent with the previous discussion of conduction capacity [22].

### 2.2. Electrochemical Test Results of Polyaniline Electrode

The micromorphology of polyaniline has great influence on its electrochemical performance. The spherical polyaniline SEM was prepared by chemical oxidation under acidic conditions (hydrochloric acid). SEM image in Figure 2a–d shows the microscopic morphology of spherical polyaniline at different polymerization times. At the initial 4 h of polymerization, polyaniline showed a disordered and staggered chain structure. Due to the slow reaction at low temperature, the polyaniline dispersed into uniform and similar spherical particles in 8 h. At 12 and 16 h, as the reaction progressed, polyaniline began to aggregate into irregular block particles [23,24].

The electrodes with different polymerization times were coated on graphite paper and named PANI-4, 8, 12, 16. Figure 3a shows the CV curve of polyaniline electrode with polymerization time. It can be seen from the image that the electrode with PANI-8 has the largest integral area. According to Nyquist curve (Figure 3b), the EIS curve of PAIN-8 has the smallest intercept with the *X*-axis, while the intercept of PANI-12 and PANI-16 is larger, which is due to the low equivalent series resistance caused by their stacked structure. Compared with other curves, the PANI-8 curve has a steeper slope, indicating that its ionic diffusion impedance is lower and it has better electrochemical performance. This corresponds to the micromorphological phenomena observed by the electron microscope. Figure 3c shows that the redox peak of the polyaniline electrode appears at 0.3 V. With the increase in scanning rate, the redox peak of the polyaniline electrode shifts to high potential, and its curve shape changes regularly, which shows that it has good discharge performance. Combined with previous work, a polyaniline electrode with excellent electrochemical properties was successfully synthesized according to the discharge law of polyaniline. In Figure 3d, an obvious discharge plateau appears at 0.3 V, which is completely consistent with the CV curve. The existence of the discharge platform proves that polyaniline undergoes a redox reaction during charging and discharging. In this process, charge transfer occurs between interfaces, which greatly increases the discharge time of the capacitor [25].

### 2.3. Demonstration of Stability and Application of Supercapacitor

In the process of acid-doping PANI, hydrogen ions combine with N atoms in the imine group to form polar delocalization, which makes PANI have high conductivity. Compared with other organic solvents, PANI was doped with sulfonic groups in MSA. The quinone ring disappeared from the molecule and the electron cloud was redistributed. The positive charge delocalization from nitrogen atom to conjugated π bond increased the conductivity of PANI. Conventional organic solvents such as DMSO and EG do not possess strong acidity and cannot be doped with PANI [26]. From Figure 4a,b, compared with EGISC and DMISC devices, the CV curves of PAIMSC and PAISC with methanesulfonic acid have obvious redox peaks, and there are also obvious discharge platforms on the GCD curves. At the scanning rate of 50 mV/s, the integral area and discharge time of the CV curve of PAIMSC are larger than those of PAISC. It also indirectly proves that MXene plays the roles of proton transport channel and charge transport carrier in the gel. The discharge time in the GCD curve can also prove the above conclusion. As shown in Figure 4c, with the increase in scanning rate, the reduction peak moves to the negative direction, which is caused by the mixed polarization of ionic resistance and electronic resistance. Even at the high scanning rate of 100 mV/s, the redox peak of the CV curve obviously appears. This indicates that the charge transfer at the interface between the electrolyte and the electrode are carried out at a high speed. The existence of the charging and discharging platform is evidence of the redox region [27]. From Figure 4d, we compare the GCD curves of different current densities. With the increase in charge–discharge current, the curve bending degree is smaller and the voltage drop is larger. When the current density increases during the charging and discharging process, the reversible redox between the electrode and the electrolyte interface will decrease, and the high current will increase the system resistance and the voltage drop. Under the current density of 0.25 mA/g, the discharge time of PAIMSC is 818.4 s, and the specific capacitance of PAIMSC reaches 204.6 F/g, which shows excellent performance among similar devices. Cyclic voltammetry was tested at different scanning rates [28].

It can be concluded from the characterization of ionogel electrolytes that PAIM-4 ionogel electrolytes have the best mechanical properties and proton conduction ability compared with other ionogel. Therefore, a sandwich-type supercapacitor was made of PAIM-4 ionogel and two tablets of polyaniline, and we have studied the electrochemical properties of the PAIMSC.

Figure 5a is cyclic voltammetry curve at a scanning rate of 20 mV/s, which is carried out at different temperatures from −20 °C to 90 °C. All the six CV curves show obvious pseudocapacitance behavior. The CV curve showed obvious pseudocapacitance, and two large oxidation–reduction peaks appeared at the corresponding potential. This is due to the presence of redox in acidic conditions, and the shape of the reaction curves varies regularly with scanning rate and temperature. With the increase in temperature, the integral area of the CV curve increased obviously, and the oxidation–reduction peaks shifted. This is because the increase in temperature promotes the charge transfer and reduces the resistance of oxidation and reduction, so it promotes the oxidation and reduction process. The oxidation potential moves to the low potential, and the reduction potential moves in the corrected direction [28,29,30].

We measured the GCD curves of the device from −20 °C to 90 °C at a current density of 1 mA/g. As the temperature rises, the discharge time of the supercapacitor becomes longer, and its IR drop decreases, so the whole platform is almost stable and regular, which has the same regularity as the CV curves (Figure 5b). All the five curves have obvious discharge platforms, which shows that PAIMSC has good specific capacitance and specific energy. With the temperature increase of 90 °C, the discharge time of the device can reach 818.4 s. When the current density is 0.25 mA/g, the energy density of PAIMSC reaching 40.9 Wh/kg when the specific capacitance of PAIMSC reaches 204.6 F/g. Therefore, the electrochemical performance is better than that of the same supercapacitor. The Nyquist curves of the same device from −20 °C to 90 °C are shown in Figure 5c. All curves are parallel to the imaginary axis in the low-frequency region, which indicates that the device has the ability of low-frequency charge transfer and ion diffusion. The intercept of each set of curves on the horizontal axis represents the equivalent series resistance. As can be seen from the diagram, the equivalent series resistance of the device decreases gradually with the increase in temperature. At 90 °C, the equivalent series resistance of the device is 2.3 Ω, which is much lower than the average supercapacitor. The slope of the low-frequency line represents the ion diffusion impedance of the electrode surface. As the temperature increases, the line approaches 90 °C, and the PAIMSC approaches pure capacitance behavior [31]. This is mainly due to the introduction of ionic liquids [VMIM][TFSI] and MXene, which increase the conductivity of the gel electrolyte and decrease the resistance of the device [30]. The characteristics of curves can all verify the theory and have concrete embodiment. The specific capacitance of the device is calculated from the discharge curves at different temperatures. The specific capacitance of the device increases with increasing temperature (Figure 5d), which matches the obtained rule and proves the ability of the device at high temperature. With the increase in working temperature, the capacitance of the supercapacitor becomes larger. However, due to the rapid transport and adsorption of electrolyte ions and the changing electrode surface, the internal resistance of supercapacitors decreases [32,33,34].

In order to verify the application and repeatability of the PAIMSC, after two PAIMSCs were connected in series and in parallel, they were tested for CV and GCD. The results of Figure 6a,b show that the CV window and maximum current of the series PAIMSCs can reach twice that of a single PAIMSC. In the series circuit, the electrochemical window increased from 1.2 V to 2.4 V. At the same time, in the parallel circuit, the CV curve area and discharge time from one PAIMSC to two PAIMSCs almost doubled. It shows that each integrated device manufactured has similar performance. In addition, by connecting these two devices in series (Figure 6d), we can light up an LED electronic watch, which shows its application prospect [35,36,37].

As shown in the illustration in Figure 6c, we connected the two ends of a device to our fingers and bent the device at different angles. When the supercapacitor was bent at this angle, we started testing the CV curve. Obviously, the shapes of the three curves were almost the same, which indicates that the PAIMSC device had a stable structure [38].

Figure 6e illustrates a Ragone plot of PAIMSC. PAIMSC exhibits higher energy densities (40.92 Wh/kg at 180 W/kg) than conventional electrochemical capacitors and other electrochemical capacitors, such as PANI/MoS2 [39], 3D-G/PANI [40], Au/PANI//AC, Au/PANI//Au/PANI [41], and PANI/HClO_4_ [42].

As can be seen from Figure 6f, PAIMSC has stable cycling performance and Cullen efficiency. At the current density of 5 mA/g, the cycle efficiency is 91.56% and the coulombic efficiency is over 93% after 10,000 cycles. On the one hand, PAIMSC at room temperature of high resistance, long-term operation will demonstrate heating phenomenon, resulting in heat loss. On the other hand, when the device is placed for a long time, a small amount of liquid will leak out. These reasons will lead to the increase in the resistance of the device and a decrease in discharge time. With the decrease in discharge time, the Coulombic efficiency and cycle efficiency decrease, according to the calculation [43].

## 3. Experimental Section

### 3.1. Materials

Titanium aluminum carbide (MAX phases), lithium fluoride (LiF), 1-vinyl-3-methylimidazolium bis(tri-fluoromethylsulfonyl)imide ([VMIM][TFSI]), ammonium persulfate (APS), Methanesulfonic Acid (MSA), Dimethyl sulfoxide (DMSO), Ethylene glycol (EG), hydrochloric acid (HCl), *N*,*N*,*N*′,*N*′-Tetramethylethylenedi-amine (TMEDA), potassium persulfate (KPS), and polyvinylidene fluoride (PVDF) were purchased from Aladdin (Shanghai, China). N-Methyl pyrrolidone (NMP), Methylene-bis-Acrylamide (MBA), acrylamide (AAm), and 1-ethyl-3-methylimidazolium bis(tri-fluoromethylsulfonyl)imide ([EMIM][TFSI]) were obtained from Macklin, Chinese Academy of Sciences. The above materials can be directly used in the experiment.

### 3.2. Preparation of MXene and PANI

Multilayer Ti_3_C_2_T_x_ was prepared by etching Ti_3_AlC_2_ powder. A total of 4 g LiF was dissolved in 100 mL 9 M HCl solution. A total of 3 g Ti_3_AlC_2_ powder was gradually added in a water bath at 40 °C for 1 d. Ti_3_C_2_T_x_ powder was obtained by washing the sediment with deionized water, filtering, and drying at 60 °C.

In order to synthesize polyaniline, 1 g aniline monomer and 120 mg APS were added to and stirred in water bath at 0 °C for 12 h. Then, the dark-green mixture was centrifuged and vacuum-dried at 60 °C to obtain the active substance in powder form. XRD images of PANI and MXene and SEM images of MXene are shown in Appendix A.

### 3.3. Preparation of PAAm/IL/MXene-X Ionogel Electrolyte

MXene powders of different masses were added to 0.5 mL MSA for ultrasonic treatment to form uniform viscous solution. A total of 0.2 g AAm, 1.7 mL [EMIM][TFSI], 0.05 mL [VMIM][TFSI], 0.3 mL MSA, 0.02 g KPS, and 0.02 g MBA were added to the reagent bottles and stirred continuously. A total of 0.02 mL TMEDA was added when there was no solid in the solution. The solution was added to the mold (400 × 100 × 2 mm^3^) and placed in an oven at 50 °C for 4 h to obtain PAAM/IL/MXene-X ionogel (PAIM-X). X is the amount (mg/mL) of MXene added: X = (0, 2, 3, 4, 5). The synthesis process and formation mechanism of PAIM-X ionogel are shown in Figure 7.

### 3.4. Preparation of PANI Electrodes Based on Graphite Paper and Supercipicator

Totals of 0.8 g PANI powder, 0.1 g graphitized carbon black, 0.1 g PVDF, and 2 mL NMP were mixed to grind and produce homogeneous oily paste. A paste of 1 × 1 cm^2^ area was coated on graphite paper. The electrode was dried at 80 °C for 24 h, and the mass of the active substance was calculated. Finally, the resulting electrode had an area of 1 cm^2^. On average, each electrode contains about 2 mg of the active substance. Due to density difference, the load of polyaniline electrode was 1.6 mg. Two obtained electrodes and PAIM ionogel were assembled into a supercapacitor (PAIMSC) for subsequent experiments and characterization. The additional gels were prepared using DMSO and EG as solvents, respectively, and then formed into supercapacitors, named DMISC and EGISC.

### 3.5. Characterization

#### 3.5.1. Proton Conductivity and Tensile Property of Electrolytes

The proton conductivity (s/cm) of the PAIM-X ionogel was calculated by Formula (1):(1)$$\sigma =\frac{R}{L\times S}$$
where σ is the ionic conductivity of PAIM ionogel in S/cm; *L* is the distance between the two electrodes in cm; *S* and *R* are the sectional area and resistance of the PAIM ionogel in cm and Ω. The gel was placed in a water bath to measure conductivity at 0–90 °C.

#### 3.5.2. Mechanical Properties of PAIM-X Ionogel Electrolyte

The PAIM-X ionogel was prepared in a cylindrical mold with a diameter of 0.5 cm and a height of 5 cm, and the mechanical properties of the gel were tested with a tensile testing machine (SHIMADZU, Model AGS-X, 100 N, Japan) at a test rate of 60 mm/min, with measurements repeated 3 times to prevent personal error.

#### 3.5.3. Electrochemical Measurements of Supercapacitors (Electrodes)

CV, EIS, GCD and long-term capacitance retention were measured on an electrochemical workstation (AUT86925, Autolab). On the basis of CV/GCD curve, the specific mass capacitance (C_s_, F/g), energy density (E, Wh/kg), and power density (P, W/kg) of PAIMSC were calculated by Formulas (2)–(4):(2)$$C_{s}=\frac{I∆t}{m∆V}$$
(3)$$E=\frac{1}{2}C_{s}∆V^{2}$$
(4)P=E∆t
where ∆t(s), I(A), ∆V(V), and m(g) represent discharge time, current, potential window during discharge, and mass of active materials on the electrode, respectively. The long-term cyclic stability of PAIMSC was tested at the current density of 10 mA/g for 10,000 cycles. The electrochemical properties of PAMSC were measured at −20–90 °C in refrigerator and water bath, respectively.

#### 3.5.4. Morphology of PANI

After gold spray treatment, the dispersion of polyaniline at different polymerization time was observed using field emission scanning electron microscope (JSM-7610F, JEOL Ltd., Tokyo, Japan).

## 4. Conclusions

To summarize, an ionogel electrolyte composed of a PAAm-[VMIM][TFSI] network, methanesulfonic acid/[EMIM][TFSI] solvent, and physical cross-linking agent MXene was successfully prepared. PAIM-X ionogel has a high conductivity of 36.4 mS/cm at 90 °C and its elastic modulus is 33.4 kPa, which is twice that of PAI ionogel. In this paper, a supercapacitor composed of ionogel and polyaniline electrode was prepared by using methanesulfonic acid/[EMIM][TFSI] as solvent for the first time. The potential window of the device is 1.2 V. When the current density is 0.3 mA/g, the specific capacitance of PAIMSC is 204.6 F/g, and the energy density is 40.92 Wh/kg. This shows that PAIMSC has a good development prospect.

## Figures and Tables

**Figure 1 molecules-28-01554-f001:**
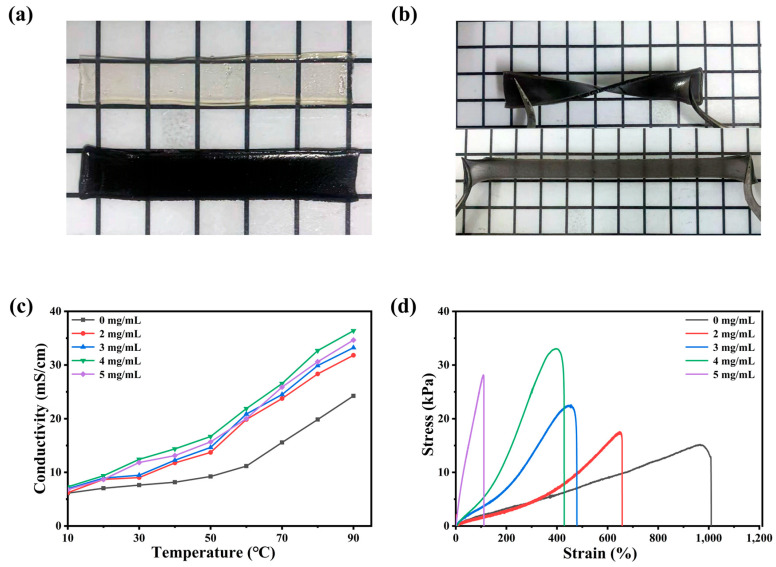
Electron photograph of ionogel. (**a**) PAIM-X ionogel and PAI ionogel; (**b**) twisted and strained PAIM-X hydrogel; (**c**) the conductivity of the different MXene content of PAIM ionogels as a function of temperature; (**d**) stress–strain curves of PAIM-X ionogel with different MXene content.

**Figure 2 molecules-28-01554-f002:**
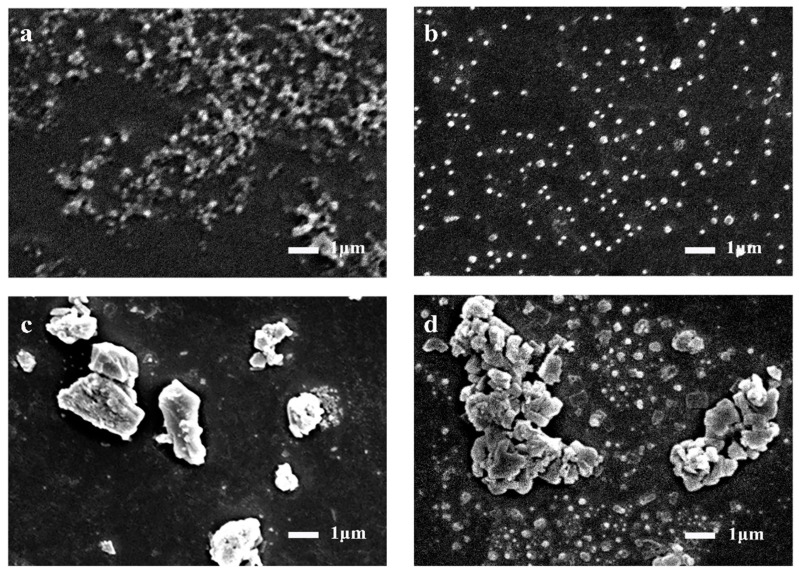
SEM of polyaniline under different polymerization times: (**a**) 4 h; (**b**) 8 h; (**c**) 12 h; (**d**) 16 h.

**Figure 3 molecules-28-01554-f003:**
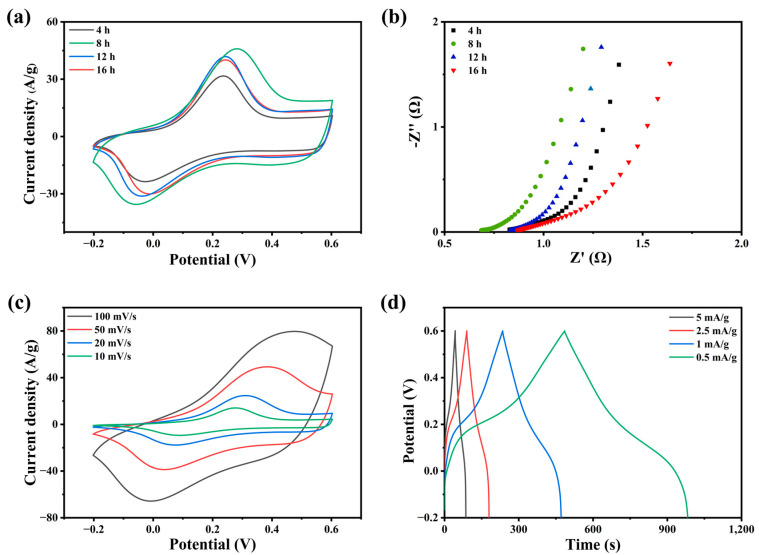
(**a**) CV curves of PANI/graphite paper electrodes with different polymerization times; (**b**) the Nyquist curves; (**c**) CV curves of PANI-8 electrode at different scanning rates; (**d**) GCD curves of PANI-8 electrode at different current densities.

**Figure 4 molecules-28-01554-f004:**
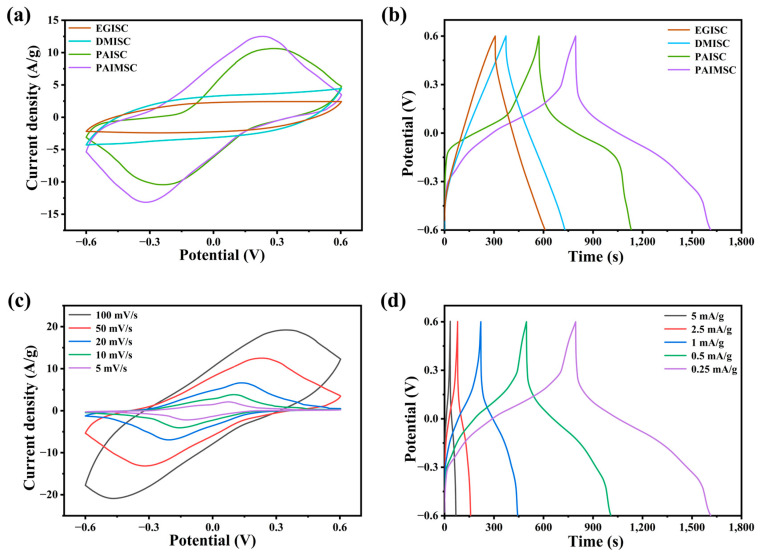
(**a**) CV curves of four devices (EGISC, DMISC, PAISC, and PAIMSC) at 50 mV/s. (**b**) GCD curves of four devices at 0.25 mA/g. (**c**) CV curves of PAIMSC devices at different scanning rates. (**d**) GCD curves of PAIMSC at different current densities.

**Figure 5 molecules-28-01554-f005:**
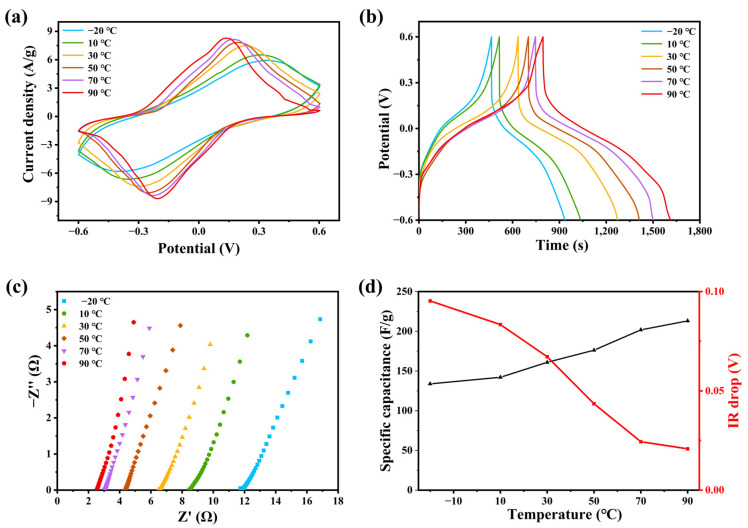
(**a**) The CV curve of PAIMSC with scanning rate of 20 mV/s at different temperatures from −20 °C to 90 °C; (**b**) GCD curves of PAIMSC at current densities of 0.25 mA/g; (**c**) different Nyquist curves of PAIMSC; (**d**) specific capacitance and IR drop at different temperatures.

**Figure 6 molecules-28-01554-f006:**
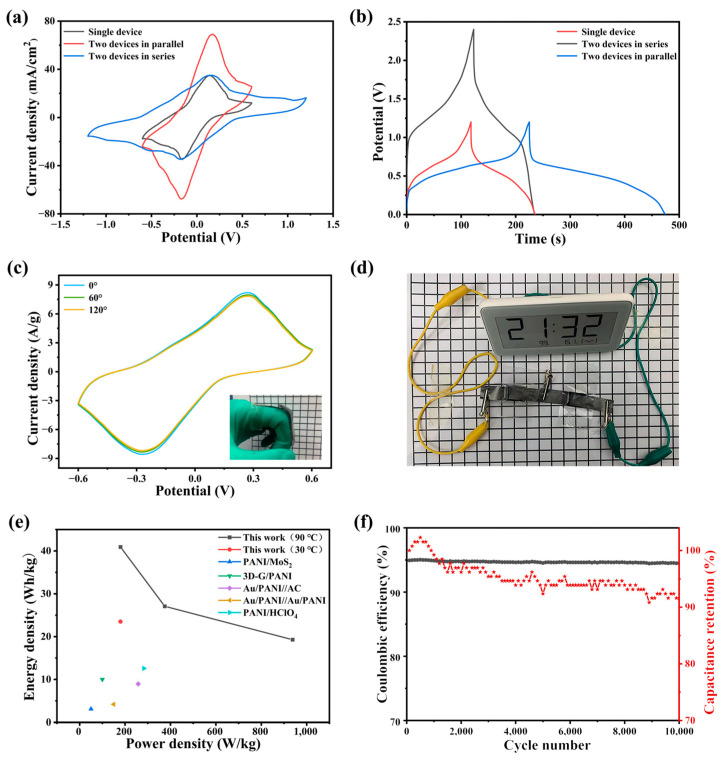
(**a**) CV curves of a single PAIMSC and two PAIMSCs in parallel or in series state; (**b**) GCD curves; (**c**) bending experiment of the PAIMSC at 100 mV/s; (**d**) physical connection diagram; (**e**) Ragone diagram of energy and power density comparison of PANI-based supercapacitors; (**f**) long-term cycling curve of PAIMSC.

**Figure 7 molecules-28-01554-f007:**
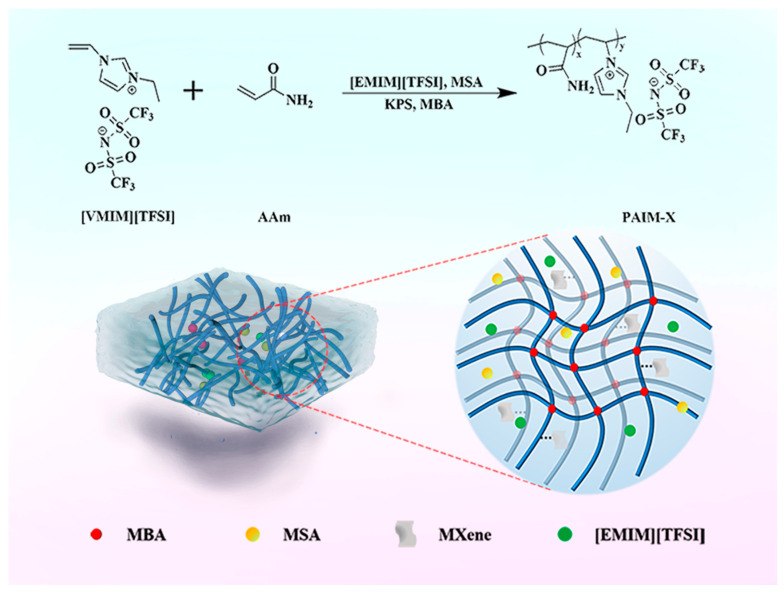
Synthesis and formation mechanism of PAIM-X ionogel.

## Data Availability

Data available on request due to restrictions e.g., privacy or ethical.

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
