# Peer review of "An All-Solid-State Flexible Supercapacitor Based on MXene/MSA Ionogel and Polyaniline Electrode with Wide Temperature Range, High Stability, and High Energy Density"

_molecules, 2023, doi:10.3390/molecules28041554_

Round 1
Reviewer 1 Report
Molecules-2145705
Major Revision
Authors have carried out synthesis of an ionogel electrolyte (PAIM-X) consisting of (VMIM) )(TFSI), PAAm, and MXene and studied solid-state supercapacitor performance. The work may be accepted upon successful implementation of the following comments.
1. Author must include basic characterisation (XRD, XPS and SEM) of MXene and PANI in supplementary information.
2. The SEM images (Figure 3) included in manuscript are of very low magnification. Please include high resolution SEM images.
3. How addition of MXene in ionogel electrolyte helps to improve the charge storage capability? Include its electrochemistry.
4. Why there is drastic change in the shape of CV and GCD curve of PAIMSC in comparison with EGISC & DMISC.
5. The device shows good stability but why there is 10% loss in columbic efficiency.? What is reason for 10% decrease in Columbic efficiency.
6. Explain in detail how ionogel used in this work help to operate the device in wide temperature -20 â—¦C to 90 â—¦C.
7. There are a few grammatical errors that need to be resolved.
Author Response
Dear Reviewer,
Thank you very much for your kind consideration of our manuscript (molecules-2145705) entitled “An all-solid-state flexible supercapacitor based on MXene/MSA ionogel and polyaniline electrode with wide temperature range, high stability and high energy density” We also appreciate the reviewers very much for their comments on our manuscript. Your comments and suggestions are very helpful for our revision. We have tried our best to revise our manuscript according to the helpful comments and highlighted in the revised version.
Please do not hesitate to contact me for any questions.
Email: wangshuang_ccut@163.com (S. Wang)
Thanks for your time and consideration again.
Sincerely yours,
Shuang Wang

Reviewer 2 Report
The research topic of the manuscript is interesting. However, the presentation quality is poor, for example, abstract, introduction. I suggested authors to do a better proofreading or reorganize the manuscript by a skilled English-speaking researcher before resubmission. I can't recommend to publish the paper at current status.
Author Response

(The authors gave the same response as above.)

Reviewer 3 Report
In this manuscript, the authors fabricate ionogel electrolytes composed of PAAm-[VMIM][TFSI] network, using MXene as physical cross-linking agent with a high conductivity.
Up to now, there are few research of all-solid supercapacitors based on ionic liquid gel electrolytes as the author mentioned. However, this manuscript has some significant shortcomings on which I would like to comment. Therefore, I do recommend to be published in Molecules after major revision.
Comment 1:
The authors give inconsistent information about the test condition. For example a stretch rate has a significant effect on stress-strain curves.
Comment 2:
In Figure 5, the author explains the discharging time of PAIMSC takes 818.4 s at 1 mA/g, however, the discharge time seems about 450 s in Figure 5d. The author should check the explained value match the graph.
Comment 3:
The authors explain the reason of low current at high current densities reflects the extremely low impedance of the device and the perfect contact between the electrode and the electrolyte. These arguments they make are not convincing to me.
Comment 4:
There is a typo overall and bold text where it shouldn’t be. Should check about it. For example, paragraph 119 and figure 2(a). Also, I recommend that authors should take more English proofreading.
Comment 5:
The authors are claiming as the reaction took place, polyaniline began to aggregate and gets bigger. But in Figure 3(b), it seems like size has decreased compared to 3(a), so it doesn’t seems like to be aggregated.
Comment 6:
The authors claim that the PAIMSC is superior to PAISC due to additional MXene materials. However, there are no reference related to the argument.
Comment 7:
The author claims that the low impedance of the PAIMSC cell is due to its perfect contact between the electrode and electrolyte. However, there is no evidence to support the results. I recommend that the authors should take the SEM image.
Author Response

(The authors gave the same response as above.)

Round 2
Reviewer 2 Report
I am ok to the updates.
Reviewer 3 Report
Thank you for your response letter. This revised manuscript is improved over the original submission. I confirmed that the comments are well reflected in the manuscript now. Therefore, I recommend to accept your manuscript to be published in Molecules.